# In vivo proximity labeling identifies cardiomyocyte protein networks during zebrafish heart regeneration

Mira I Pronobis[1,2], Susan Zheng[1], Sumeet Pal Singh[3], Joseph A Goldman[4], Kenneth D Poss[1,2]*

[1]Department of Cell Biology, Duke University Medical Center, Durham, United States; [2]Regeneration Next, Duke University, Durham, United States; [3]IRIBHM, Université Libre de Bruxelles (ULB), Brussels, Belgium; [4]Department of Biological Chemistry and Pharmacology, The Ohio State University Medical Center, Columbus, United States

**Abstract** Strategies have not been available until recently to uncover interacting protein networks specific to key cell types, their subcellular compartments, and their major regulators during complex in vivo events. Here, we apply BioID2 proximity labeling to capture protein networks acting within cardiomyocytes during a key model of innate heart regeneration in zebrafish. Transgenic zebrafish expressing a promiscuous BirA2 localized to the entire myocardial cell or membrane compartment were generated, each identifying distinct proteomes in adult cardiomyocytes that became altered during regeneration. BioID2 profiling for interactors with ErbB2, a co-receptor for the cardiomyocyte mitogen Nrg1, implicated Rho A as a target of ErbB2 signaling in cardiomyocytes. Blockade of Rho A during heart regeneration, or during cardiogenic stimulation by the mitogenic influences Nrg1, Vegfaa, or vitamin D, disrupted muscle creation. Our findings reveal proximity labeling as a useful resource to interrogate cell proteomes and signaling networks during tissue regeneration in zebrafish.

*For correspondence:
Kenneth.Poss@duke.edu

**Competing interests:** The authors declare that no competing interests exist.

## Introduction

Understanding how and why tissue regeneration occurs is a central objective of developmental biology. Genome-wide transcriptome or chromatin profiling experiments using RNA-seq, scRNA-seq, ATAC-seq, or ChIP-seq approaches have helped to identify novel factors, mechanisms, and concepts that have informed our understanding of regeneration in many tissues, injury contexts, and species (*Goldman et al., 2017*; *Hoang et al., 2020*; *Johnson et al., 2020*). Although key for detecting and inferring changes in gene transcription, chromatin organization, or DNA–protein associations, these strategies are limited in their abilities to identify protein and signaling complexes that carry out changes at the cell and tissue level. Co-immunoprecipitations and affinity-purification mass spectrometry (AP/MS/MS) are well-established techniques to probe protein interactions and networks. Yet, they require tissue dissociation and flow cytometry to acquire cell type-specific interactions, and weak interactions between proteins often escape detection (*Perkins et al., 2010*).

Proximity labeling is a relatively recent technology established to probe protein interactions in specific sub-tissue and subcellular compartments, represented by multiple versions including BioID2, TurboID, and APEX2 (*Kim et al., 2016*; *Branon et al., 2018*; *Hung et al., 2016*). Here, engineered ligases that attach biotin to lysine residues are fused with a protein of interest such that, when expressed in cells, proteins in close proximity are indiscriminately biotinylated. The biotinylated proteome is isolated by streptavidin affinity, and proteins are identified by quantitative mass spectrometry. Most studies have applied proximity labeling to in vitro tissue culture systems, whereas many

fewer have explored in vivo networks in mice or *Drosophila* (*Spence et al., 2019*; *Feng et al., 2020*; *Rudolph et al., 2020*; *Mannix et al., 2019*; *Cho et al., 2020*). In the latter studies, proximity labeling captured the proteome within interneuron–neuron interactions, within muscle sarcomeres, and in the ring canal of *Drosophila* oocytes, producing new insights into protein network assemblies and signaling. The principles of proximity labeling should be capable of identifying key protein networks during dynamic cellular processes like embryonic development and tissue regeneration without the need for tissue disruption. Notably, proximity labeling has yet to be applied in central animal models of regeneration such as zebrafish and axolotl salamanders.

Unlike mammals, teleost zebrafish regenerate cardiac muscle after injury with minimal scar formation, based on division of pre-existing cardiomyocytes (*Poss et al., 2002*; *Kikuchi et al., 2010*; *Jopling et al., 2010*). Understanding mechanisms of innate cardiac repair in animals like zebrafish has the potential to define and overcome barriers to cardiac regeneration in humans. Evidence to date indicates that heart regeneration in zebrafish is a relentless process that outcompetes scarring to recover the heart function, rather than a linear series of molecular steps and synchronized cellular progressions. Intrinsic programs in cardiomyocytes and the ability to receive extrinsic proliferative cues are prime cardiomyocytes for cell division. Many ligands, receptors, and transcription factors have been shown to be required for heart regeneration (*Xiao et al., 2016*; *Zhao et al., 2019*; *Dogra et al., 2017*; *Chablais and Jazwinska, 2012*; *Wu et al., 2016*; *Pronobis and Poss, 2020*). Three secreted factors, Nrg1, Vegfaa, and vitamin D, have been demonstrated to have instructive mitogenic effects, characterized by the ability to induce cardiomyocyte proliferation in zebrafish in the absence of injury (*Gemberling et al., 2015*; *Karra et al., 2018*; *Han et al., 2019*). Nrg1/ErbB2 signaling can stimulate cardiomyocyte division in adult mice, most notably when ErbB2 is experimentally expressed in the form of a ligand-independent activator (*Bersell et al., 2009*; *D'Uva et al., 2015*; *Gordon et al., 2009*). ErbB2 signaling was recently reported to work in part through regulation of YAP, a transcriptional activator that can impact cytoskeletal remodeling and cell division in cardiomyocytes during cardiac growth or repair in mice (*Leach et al., 2017*; *Aharonov et al., 2020*; *Heallen et al., 2013*; *Heallen et al., 2011*). In zebrafish, Nrg1/ErbB2 signals in part by promoting glycolysis versus mitochondrial oxidative phosphorylation and reducing Tp53 levels through upregulation of its inhibitor Mdm2 (*Shoffner et al., 2020*; *Honkoop et al., 2019*; *Fukuda et al., 2019*).

Signal transduction mechanisms by which ErbB2 and other mitogenic factors support cardiomyocyte proliferation versus other outcomes could be illuminated by customized proteomic profiling. Proteomic studies of heart regeneration to date have faced challenges, including lack of cell-type specificity and limited detection of changes in poorly represented members of the proteome due to the abundance of sarcomeric proteins (*Wang et al., 2013*; *Ma et al., 2018*; *Garcia-Puig et al., 2019*). Here, we applied a transgenic BioID2 strategy to capture cell-specific proteome changes in cardiomyocytes during heart regeneration in adult zebrafish. We show that BioID2 can be used to track changes in protein levels in whole cardiomyocytes or their membrane compartments during regeneration. We also investigate the proximal proteome of ErbB2 in cardiomyocytes, finding an increase in association with the small GTPase Rho A during regeneration, a protein with activity we find to be essential for injury-induced regeneration and the proliferative response to mitogens. Our findings demonstrate the utility of proximity labeling as a resource to interrogate cell proteomes and signaling networks during tissue regeneration in zebrafish, and they identify a key role of small GTPases in injury-induced cardiogenesis.

## Results and discussion

### BioID2 detects cardiomyocyte proteomes in adult zebrafish

BioID2 employs the promiscuous BirA2, which attaches biotin molecules to the lysines of proteins within an approximate distance of 10 nm (*Kim et al., 2016*). We reasoned that the lysines of sarcomeric proteins might be less accessible when packed in supermolecular complexes, and therefore less likely to undergo biotinylation. To determine whether BioID2 is functional in cardiomyocytes of adult zebrafish, we generated a transgenic line with a cassette encoding BirA2 fused to GFP and placed downstream of the *myosin light chain 7* (*cmlc2*, or *mlc7*) promoter (*Figure 1A, B*). BirA2-GFP was readily detected in adult zebrafish cardiac chambers, where it was distributed throughout the cytoplasm and nuclei of adult cardiomyocytes (*Figure 1B*). To test whether BirA2 expressed in this

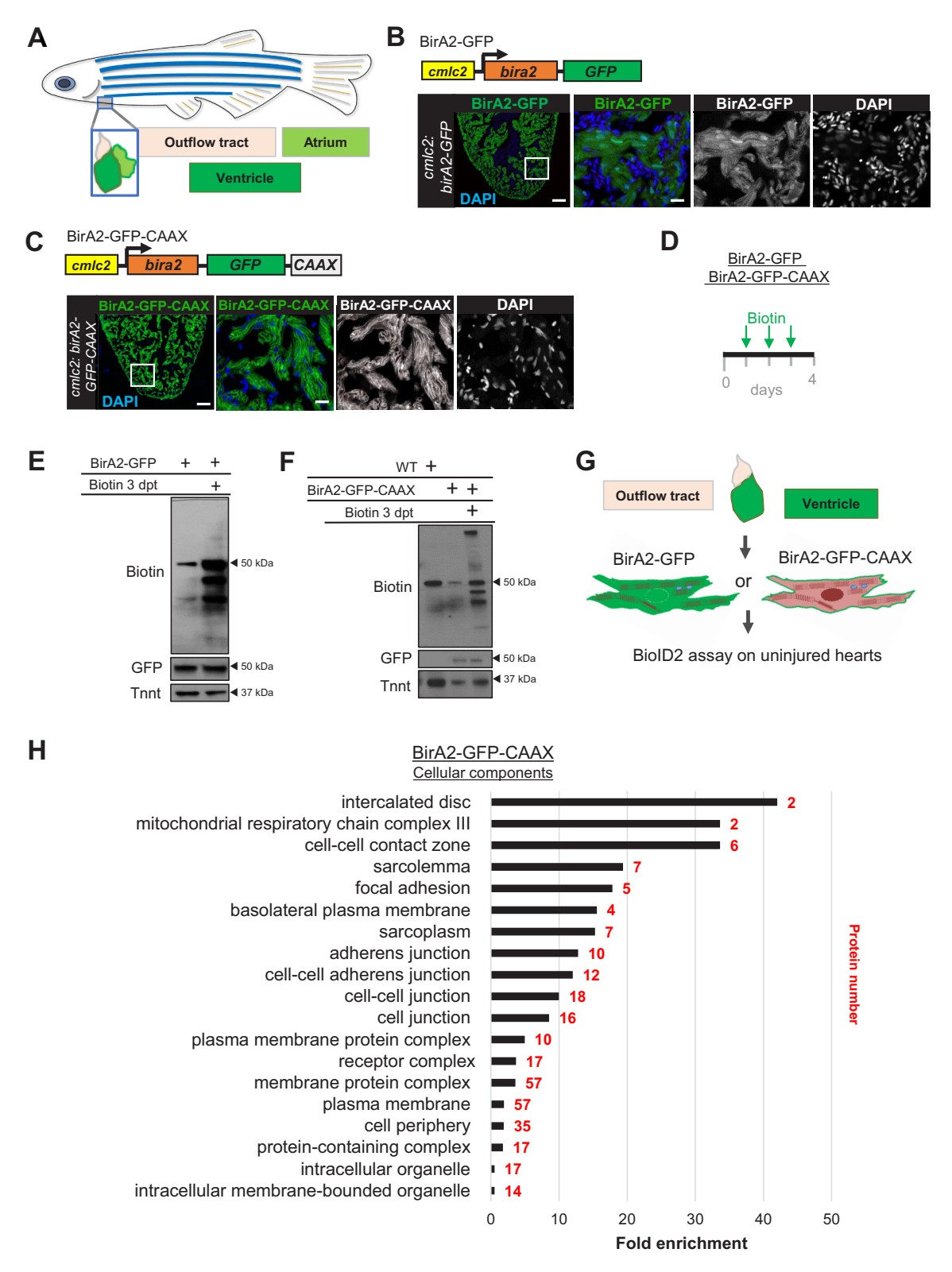

**Figure 1.** In vivo BioID2 captures the proteome in cardiomyocytes of adult zebrafish hearts. (**A**) Schematic representation of a zebrafish heart. The heart consists of a ventricle, atrium, and outflow tract. (**B**) BirA2-GFP is expressed in cardiomyocytes via the *cmlc2* promoter. Section of ventricle from *cmlc2: birA2-GFP* transgenic animal. BirA2 is distributed throughout the cardiomyocytes, including the nucleus. DAPI staining and native GFP signal are shown. (**C**) BirA2-GFP-CAAX is expressed in cardiomyocytes via the *cmlc2* promoter. Section of ventricle from *cmlc2:bira2-GFP-CAAX* transgenic

*Figure 1 continued on next page*

*Figure 1 continued*

animal. The CAAX-tag localizes BirA2 to the membranes of cardiomyocytes. (**D**) Timeline of biotin administration by intraperitoneal injections (IP injections). (**E**) Western blot analysis of biotinylation activity of BirA2-GFP-expressing ventricles. BirA2 is functional and biotinylates sufficiently after three daily biotin injections. Endogenous biotinylated carboxylase was detected in untreated and biotin-treated hearts. (**F**) Western blot analysis of BirA2-GFP-CAAX-expressing ventricles. BirA2 is functional and biotinylates sufficiently after three daily injections with biotin. WT: wild type. (**G**) Schematic summary of BioID2 assay on uninjured zebrafish hearts. (**H**) Over-representation test for cellular components. BioID for BirA-GFP-CAAX enriches for membrane-associated proteins. 343 total proteins were gated at a 2.5-fold change when normalized to BirA-GFP. p<0.001, false discovery rate (FDR) < 0.015%. dpt: days post treatment. Scale bar in images, 50 μm; magnification scale bar: 10 μm.

The online version of this article includes the following figure supplement(s) for figure 1:

**Figure supplement 1.** Evaluation of biotinylation in BirA2-GFP-expressing transgenic zebrafish.
**Figure supplement 2.** Evaluation of BirA2-GFP-CAAX localization and biotinylation.

manner biotinylates the cardiomyocyte proteome, we supplied biotin either in the aquarium water or via intraperitoneal injections over three consecutive days. We found that effective protein biotinylation occurred only when biotin was delivered by injection (*Figure 1D, E*, *Figure 1—figure supplement 1A–E*). Thus, transgenic BioID2 systems can biotinylate proteins in cardiomyocytes of adult zebrafish.

To test whether BioID2 can detect proteins that are compartmentalized in cardiomyocytes, we generated transgenic zebrafish with BirA2-GFP expected to be localized to membranes via a CAAX motif (*Figure 1A, C*). We found that the majority of BirA2-GFP-CAAX signal associated with the cardiomyocyte plasma membrane in *cmlc2:birA2-GFP-CAAX* hearts (*Figure 1C*, *Figure 1—figure supplement 2A, B*). Although BirA2-GFP-CAAX was enriched at membranes, direct staining for biotin revealed signals at membranes and cytosol, suggesting release of labeled proteins from the membrane over the 3-day incubation period (*Figure 1D–F*, *Figure 1—figure supplement 2C–G*). To capture the membrane proteome of cardiomyocytes, we conducted the BioID2 assay on adult *cmlc2:bira2-GFP* or *cmlc2:bira2-GFP-CAAX* hearts (*Figure 1G*). Quantitative BioID2 analysis captured 1113 distinct proteins, of which 343 proteins showed a 2.5-fold enrichment in BirA2-CAAX samples when compared to BirA2-GFP samples (*Supplementary file 1*). The threshold of 2.5-fold has previously been used in in vivo BioID2 studies in mice (*Uezu et al., 2016*). Gene ontology analysis for cellular components of the 343 proteins revealed enrichment for proteins associated with cellular junctions, plasma membrane, and membrane-bound organelles (*Figure 1H*). Taken together, our results indicate that in vivo proximity labeling using BioID2 is functional in zebrafish and can profile proteomes of highly specialized cells like cardiomyocytes.

## BioID2 detects proteome changes in cardiomyocytes during heart regeneration

To capture the changes in the protein network of cardiomyocytes during regeneration, we combined our BirA2 system with a transgenic system to ablate ~60% of cardiomyocytes through the tamoxifen (4-HT)-inducible expression of a Cre-induced *diphtheria toxin A* gene (*Wang et al., 2011*). We injected biotin for 3 days and collected hearts at 14 days post incubation (dpi), followed by purification of the biotinylated proteome (*Figure 2A, B*). Proliferation of cardiomyocytes in this injury model is most prominent around 7 dpi, with regeneration of the myocardium essentially complete by 28 dpi (*Figure 2—figure supplement 1A–C*). We assessed *cmlc2:bira2-GFP* hearts in two different conditions, uninjured (treated with 4-HT but no *CreER* transgene) and 14 dpi. Quantitative mass spectrometry analysis of pooled samples revealed that 208 proteins showed a 1.5-fold or greater change during regeneration when compared to uninjured hearts (*Figure 2D*, *Supplementary file 2*).

It is common in proximity labeling assays to normalize levels of the protein of interest to those of a diffuse BirA2-labeled protein. However, this requires a high threshold for protein changes to avoid false positives, diminishing the gating of proteins with subtle fold changes. Normalizing the regenerating heart data set of a transgenic strain to the uninjured control of the same strain allowed a low threshold. Gene ontology analysis (over-representation test) for biological processes characterized the proteins with increased levels during regeneration in the whole-cell BioID2 set as involved in wound healing, cardiac development, muscle cell differentiation, and response to wounding, whereas decreased proteins were enriched for metabolites and energy precursors (*Figure 2D*). The latter category includes the generation of ADP to ATP and oxidative phosphorylation, implying that

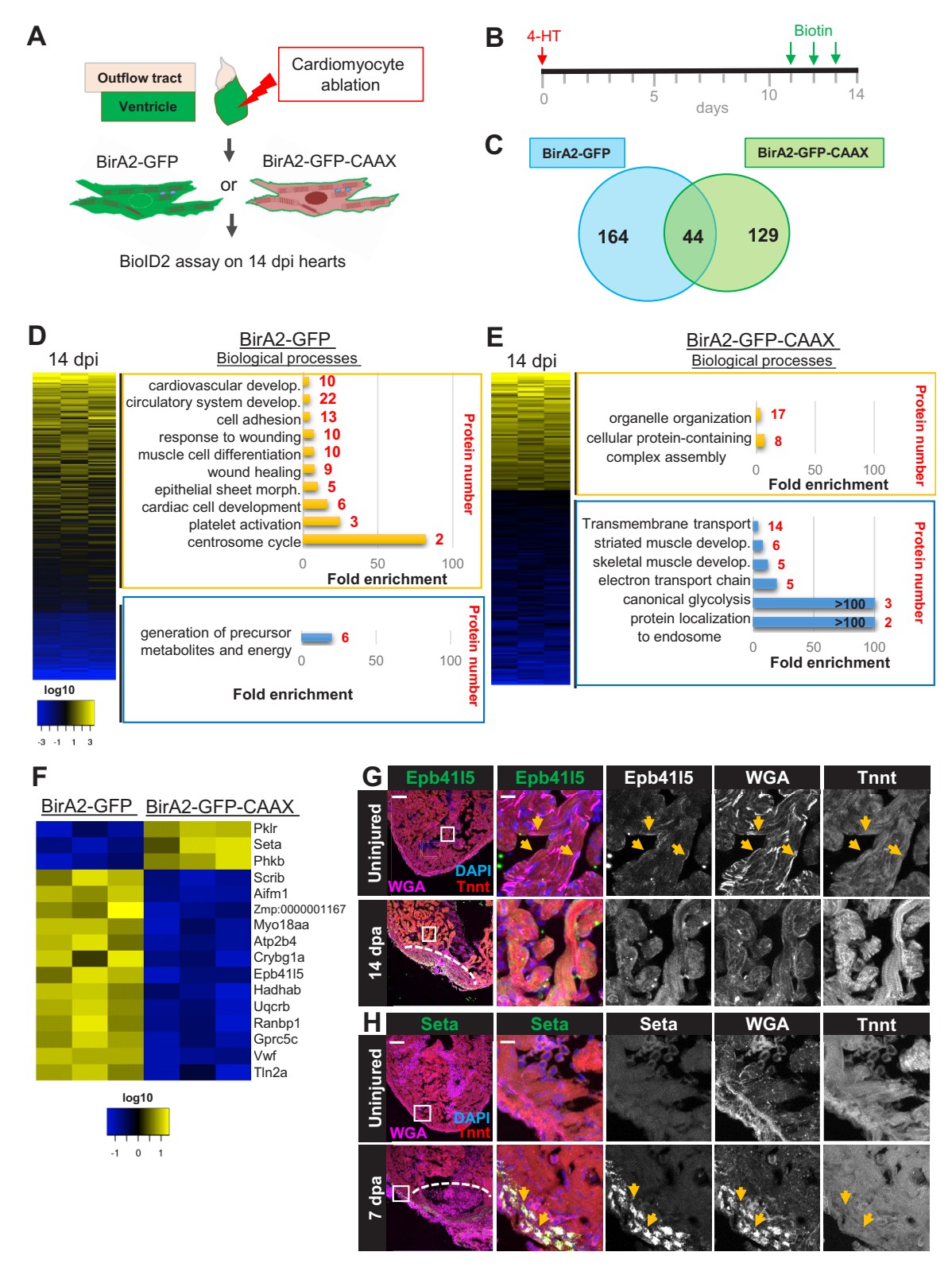

**Figure 2.** BioID2 identifies changes in membrane protein levels during heart regeneration. (**A**) Schematic overview of experimental workflow. *cmlc2: birA2-GFP* or *cmlc2:bira2-GFP-CAAX* ventricles were collected as uninjured samples or 14 days after induced cardiomyocyte ablation (dpi). (**B**) Timeline of injury and biotin injections. (**C**) Venn diagram comparing proteins captured by either the whole-cell BioID2 assay or the membrane BioID2 data set that display at least a 1.5-fold change. (**D**) Summary of BioID2 analysis of *cmlc2:birA2-GFP* hearts during regeneration. Left: heat map of proteins found

*Figure 2 continued on next page*

*Figure 2 continued*

in triplicates of quantitative mass spectrometry analysis. 208 proteins displayed a 1.5-fold change during regeneration when compared to uninjured hearts (p<0.05). Of these, most protein levels increase during heart regeneration. Protein level changes in log10 scale. Right: gene ontology analysis of BirA2-GFP BioID2 data set. Over-representation test of biological processes for at least 1.5-fold increased (yellow) and at least 1.5-fold decreased proteins (blue). Fold enrichment is shown, and protein number is indicated in red. p<0.001, FDR < 0.04%. (E) Summary of BioID2 on *cmlc2:bira2-GFP-CAAX* hearts. Left: heat map of proteins found in triplicates of quantitative mass spectrometry analysis. 173 proteins displayed an at least 1.5-fold change when compared to uninjured hearts (p<0.05). Of these, most proteins decreased at the membrane during heart regeneration. Right: gene ontology analysis of BirA2-GFP-CAAX BioID2 data set. p<0.001, FDR < 0.05%. (F) Heat map of proteins that have been identified in the BioID2 whole-cell and membrane data sets and that display opposing changes in levels. Those shown changed at least 1.5-fold, p<0.05. Heat map summarizes fold changes measured in three separate pooled samples. (G) Immunofluorescence against indicated proteins in ventricles. Epb41l5 is localized to the plasma membrane (marked by wheat germ agglutinin [WGA] staining) in uninjured hearts, with cytoplasmic fluorescence signals increasing during regeneration of resected tissue (14 days post amputation [dpa]). (H) Seta is poorly detected in uninjured hearts, rising 7 days after resection injury in the epicardium and compact layer of the heart. Seta localizes around the nucleus and colocalizes with the membrane marker WGA. Scale bar in images, 50 μm; magnification scale bar: 10 μm.

The online version of this article includes the following figure supplement(s) for figure 2:

**Figure supplement 1.** Cardiomyocyte ablation in *cmlc2:birA2-GFP* fish, and summary of proteins that display largest changes in levels.

**Figure supplement 2.** Comparison of the BioID2 data set with published transcriptome or epigenome data sets.

**Figure supplement 3.** Cardiomyocyte ablation in *cmlc2:bira2-GFP-CAAX* fish, and summary of proteins that display largest changes levels.

**Figure supplement 4.** Protein levels and localization from BioID2 comparison of whole cardiomyocytes and membranes.

these processes are dampened during dedifferentiation and proliferation of cardiomyocytes, as suggested by recent publications (*Honkoop et al., 2019*; *Fukuda et al., 2019*). The top 10% of increased proteins included ErbB2 and the mTOR signaling protein Rictora (*Figure 2—figure supplement 1D*), each previously implicated in heart regeneration (*Gemberling et al., 2015*; *Sciarretta et al., 2015*). We compared the BioID2 data set to published RNA-seq or ChIP-seq data sets, which each identified a much greater number of differential products than BioID2 (*Goldman et al., 2017*; *Rudolph et al., 2020*; *Kang et al., 2016*; *Ben-Yair et al., 2019*). Correlations were generally weak, with the highest being an RNA-seq data set from 14 dpi hearts at r = 0.39 (*Figure 2—figure supplement 2A–D*; *Kang et al., 2016*). We suspect that any post-transcriptional regulation incorporated in BioID2 data contributes to these weak correlations. Beyond this, there are several possible technical explanations, including differences in assessed timepoints, injury models, and cell types, and limitations of BioID2 in capturing targets. Regardless, our results indicate that proximity labeling with BirA2 can identify biological processes that are involved in tissue regeneration and proteins that have been reported to regulate cardiac repair.

To investigate how the membrane proteome changes after heart injury, we examined *cmlc2: bira2-GFP-CAAX* samples at 14 dpi (*Figure 2A, B*, *Figure 2—figure supplement 3A–C*). We identified 173 proteins with a 1.5-fold or greater change in membrane-associated proteins during regeneration when normalized to uninjured hearts (*Figure 2C, E*, *Supplementary file 3*), with the majority of these proteins reducing presence during regeneration. This contrasts with BirA2-GFP samples, which showed more proteins increasing levels during regeneration than those decreasing (*Figure 2D* vs. *Figure 2E*). Proteins with reduced membrane levels during regeneration showed over-representation of involvement in transmembrane transport, muscle development, and energy production (electron transport and glycolysis), while proteins increased at the membrane during regeneration were involved in protein complex assembly and organelle organization (*Figure 2E*). Intracellular restructuring is known to occur in dividing cardiomyocytes (*Uribe et al., 2018*). Upon comparison of the top and bottom 10% of the membrane profiling data with BirA2-GFP data (*Figure 2—figure supplement 3D* vs. *Figure 2—figure supplement 1D*), very few proteins were found in both data sets, suggesting specificity of the membrane BioID2 data.

Comparison of proteins found in BirA2-GFP and BirA2-GFP-CAAX data sets identified 44 shared proteins (*Figure 2C*). Of these, 16 proteins showed opposing changes in levels at the membrane or in the whole cell (*Figure 2F*). To visualize dynamism of proteins for which there were available candidate antibodies during regeneration, we assessed some of these proteins by immunofluorescence in a model of regeneration in which ~20% of the ventricle is resected (*Poss et al., 2002*). Erythrocyte membrane protein band 4.1 like 5 (Epb41l5) is involved in animal organ development and has been implicated as a positive regulator of Notch signaling, a pathway involved in heart regeneration

(*Matsuda et al., 2016*; *Zhao et al., 2014*, *Figure 2F, G*). Epb41l5 was predominantly localized at the plasma membrane (*Figure 2G*, arrows) and increased presence throughout cardiomyocytes during regeneration, consistent with the BioID2 data. Talin 2, an adapter protein that couples integrin focal adhesions to the actin cytoskeleton (*Figure 2—figure supplement 4A, B*), displayed low levels in uninjured hearts (*Figure 2—figure supplement 4B*). Upon injury, Talin 2 levels increased near the wound site as well as in ventricular muscle away from the injury, with a punctate pattern appearing that resembled focal adhesions (*Figure 2—figure supplement 4B*). These results are consistent with a remodeling of focal adhesions during myocardial stress induction that has been reported for Talin 1 (*Manso et al., 2013*). To assess BioID2 results implicating proteins that increase levels at the cell membrane during regeneration, we identified antibodies against SET nuclear proto-oncogene a (Seta), which is involved in nucleosome assembly (*Figure 2F, H*, *Figure 2—figure supplement 4A, C*). Seta levels were undetectable in uninjured hearts, yet became prominent in compact muscle and epicardial tissue in injured ventricles (*Figure 2H*, *Figure 2—figure supplement 4C*). The Seta signal was perinuclear and colocalized with wheat germ agglutinin (WGA), which indicates that Seta associates in part with the nuclear membrane and endomembranes after heart injury (*Figure 2H*, arrows, *Figure 2—figure supplement 4C*). As Seta has been proposed to silence histone acetyl-transferase-dependent transcription, its recruitment to nuclear and endomembranes could impact this function (*Seo et al., 2001*). Taken together, our data indicate that BioID2 can capture changes in the proteomes of cardiomyocytes during heart regeneration, providing a resource for candidate effectors.

## Rho A interacts with ErbB2 during heart regeneration

Nrg1 ligand has been reported to bind its receptor ErbB4 on cardiomyocytes, causing heterodimerization of Erbb4 and its co-receptor ErbB2 (*Bersell et al., 2009*; *Harskamp et al., 2016*). To probe the network of ErbB2 signaling during heart regeneration in zebrafish, we generated the transgenic line *cmlc2:erbb2-birA2-HA-P2A-GFP* (*Figure 3A*), with BirA2 fused to the C-terminus of ErbB2. We detected ErbB2-BirA2-HA via western blotting analysis, although we could not visualize it by immunofluorescence (*Figure 3—figure supplement 1A*). We also detected biotinylation of cardiac proteins in this line in the presence of injected biotin (*Figure 3—figure supplement 1A*). To assess dynamic ErbB2 networks during heart regeneration, we crossed *cmlc2:erbb2-birA2-HA-P2A-GFP* with Z-CAT animals to ablate cardiomyocytes as described above and collected ventricles at 14 dpi (*Figure 3B*, *Figure 3—figure supplement 1B–D*). Quantitative mass spectrometry analysis identified 667 proteins, of which 108 showed a >1.5-fold change during regeneration when normalized to uninjured control hearts (*Supplementary file 4*). Using STRING and BioGRID databases, we identified nine known ErbB2 interactors in our data set. Three of these proteins showed increased association with ErbB2 during regeneration, whereas two had decreased association (*Figure 3C*, yellow and blue circles, respectively). Next, we examined the levels of these candidates as reported in published transcriptome and proteome data sets (*Goldman et al., 2017*; *Wang et al., 2013*; *Ma et al., 2018*; *Kang et al., 2016*; *Ben-Yair et al., 2019*), and found that overall expression levels were not predictive of ErbB2 association (*Supplementary file 5*). ErbB4, encoded by *erbb4a* and *b* paralogs, was not detected in any of the BioID2 data sets, which might either indicate that ErbB4 does not associate with ErbB2 in the zebrafish heart, or that their C-terminal regions are not in close proximity, or a limitation in sensitivity of the assays. The strongest detected increase was an association with the Rho GTPase Rho A-b, and pathway analyses indicated enrichment for proteins involved in cytoskeletal regulation by Rho GTPases (*Figure 3C, D*). Rho A activation by ErbB2 has been previously reported to promote pro-metastatic and invasive behavior of human breast cancer cells and contribute to mammary adenocarcinoma tumorigenesis (*Brantley-Sieders et al., 2008*; *Worzfeld et al., 2012*). Moreover, a recent study implicated Rho A mislocalization with aberrant spindle orientation leading to binucleation, a common feature of mammalian CMs that is thought to inhibit cardiomyocytes (CM) proliferation (*Leone et al., 2018*). To test whether ErbB2-Rho A association is increased during regeneration, we conducted co-immunoprecipitations (*Figure 3E*). Interestingly, Rho A was pulled down with ErbB2-BirA2-HA in regenerating heart samples, but not from uninjured myocardium, suggesting a context-dependent association (*Figure 3E*). Due to their high conservation, we were unable to identify which of the Rho A proteins, Rho A-a, Rho A-b, or Rho A-c, increased its association with ErbB2 (*Figure 3—figure supplement 2*).

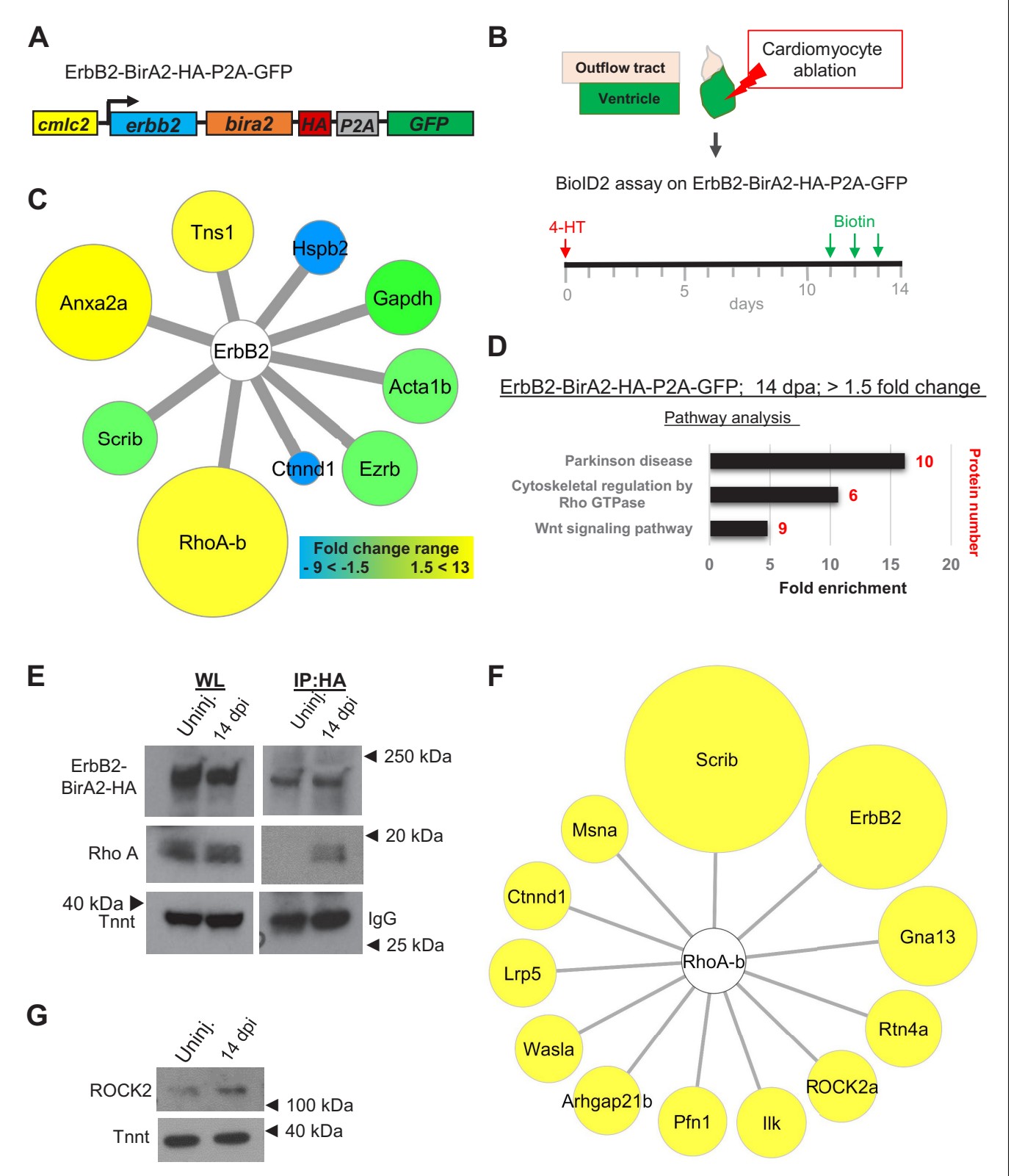

**Figure 3.** BioID2 identifies Rho A as a downstream target of ErbB2. (**A**) ErbB2-BirA2-HA-P2A-GFP is expressed in cardiomyocytes via the *cmlc2* promoter. (**B**) Summary of experiment and timeline. *cmlc2:erbb2-birA2-HA-P2A-GFP* ventricles were collected as uninjured samples or 14 days after induced cardiomyocyte ablation (dpi). (**C**) Known direct interactors of ErbB2 that were captured in the ErbB2 BioID2 assay. 108 proteins showed a change >1.5-fold when normalized to uninjured *cmlc2:erbb2-birA2-HA-P2A-GFP* ventricles. These data were analyzed for known ErbB2 interactors.
*Figure 3 continued on next page*

Figure 3 continued

Colors and size of interactor correspond to fold changes identified during regeneration. Green: no change; blue: levels decrease; yellow: levels increase. Tns1: tensin 1 (1.57-fold); Hspb2: heat shock protein beta 2 (−2.43-fold); Gapdh: glyceraldehyde 3-phosphate dehydrogenase (p>0.05); Acta1b: actin alpha 1 (p>0.05); Ezrb: ezrin b (p>0.05); Ctnnd1: catenin δ1 (−4.59-fold); RhoA-b: Rho A-b (13-fold); Scrib: scribble (p>0.05); Anxa2a: annexin A2a (7.31-fold). (D) Over-representation test – pathway analysis of proteins increased 1.5-fold or more in ErbB2-BirA data set. p<0.001, FDR < 0.005%. Fold enrichment is shown, and protein number in red. (E) Co-immunoprecipitation of ErbB2-BirA2-HA from uninjured or regenerating *cmlc2:erbb2-birA2-HA-P2A-GFP* hearts. Rho A association with ErbB2 is increased after injury. Anti-HA antibody was used for ErbB2 detection, and troponin T and IgG were used as loading controls. (F) Analysis of known Rho A interactors in BirA2-GFP BioID2 data set. All known direct interactors were found to be increased when normalized to uninjured hearts. Size of circles indicates fold change; proteins are sorted clockwise after fold change from high to low. Scrib: scribble (90.9-fold); ErbB2: Erb-b2 receptor tyrosine kinase 2 (52.9-fold); Gna13: guanine nucleotide binding protein alpha 13a (25.8-fold); Rtn4a: reticulon 4a (8.82-fold); ROCK2a: Rho-associated, coiled-coil containing protein kinase 2a (5.05-fold); Ilk: integrin-linked kinase (3.91-fold); Pfn1: profilin 1 (3.25-fold); Arhgap21b: Rho GTPase activating protein 21b (2.65-fold); Wasla: WASP-like actin nucleation promoting factor a (2.5-fold); Lrp5: low-density lipoprotein receptor-related protein 5 (2.21-fold); Ctnnd1: catenin δ1 (2.2-fold); Msna: moesin a (1.9-fold). (G) Western blot analysis of ROCK2 levels in uninjured and regenerating hearts. ROCK2 levels are increased during heart regeneration.

The online version of this article includes the following figure supplement(s) for figure 3:

**Figure supplement 1.** Biotinylation in *cmlc2:erbb2-birA2-HA-P2A-GFP* ventricles.
**Figure supplement 2.** Alignment of Rho A antibody immunogen with Rho A proteins.

As many direct interactors of Rho A have been identified, we searched our data set of whole cardiomyocyte changes at 14 dpi for these proteins. Levels of myocardial Rho A proteins detected by BioID2 were similar in uninjured and regenerating hearts. Yet, all known Rho A interactors detected in our *cmlc2:birA2-GFP* data set showed elevated levels after injury, including ErbB2 (59.2-fold; *Figure 3F*). Most of the detected Rho A interactors are involved in cytoskeletal organization such as moesin a, WASP-like actin nucleation promoting factor a, profilin 1, catenin δ1, and ROCK2a, or cell adhesion such as scribble and integrin-linked kinase (*Figure 3F*). The signal transducer Rho GTPase activating protein 21b, the guanine nucleotide binding protein (G protein) alpha 13a, which is involved in the formation of heart fields, and reticulon 4a, which plays a role in epithelial mesenchymal transition (EMT) transition, were also found to be increased after injury in the whole-cell BioID2 data set (*Figure 3F*, *Ye et al., 2015*; *Zhao et al., 2015*). Rho A signaling is known to facilitate cytoskeletal changes via Rho-associated protein kinases (ROCK), which have been reported to regulate the DNA binding activity of the cardiogenic transcription factor Gata4 and nuclear localization of SRF in cell culture (*Kaarbo et al., 2013*; *Liu et al., 2003*; *Yanazume et al., 2002*). ROCK2a levels were elevated in our whole-cell BioID2 data set, further confirmed by western blotting (*Figure 3G*), consistent with the idea that ErbB2–Rho A association could impact ROCK2 levels in cardiomyocytes after injury. Thus, BioID2 interrogation of the co-receptor ErbB2 captured the network of ErbB2 during heart regeneration, a potential resource to uncover novel ErbB2-interacting proteins, and revealed increased association of Rho A and ErbB2 in regenerating cardiomyocytes.

## Rho A activity is essential for injury- and mitogen-stimulated cardiomyocyte proliferation

To evaluate a potential role for Rho A signaling during cardiac repair, we treated zebrafish with the Rho A inhibitor Rhosin for three consecutive days after partial ventricular resection and assessed indicators of cardiomyocyte proliferation (*Figure 4A, B*; *Miranda-Rodríguez et al., 2017*). In these experiments, Rhosin reduced the proliferation index by ~68% (*Figure 4C*). To test whether regeneration is inhibited or delayed when Rho A signaling is blocked, we generated a transgenic line that enables tamoxifen-inducible expression of a dominant-negative Rho A cassette in cardiomyocytes. (*Figure 4F*, top panel). In animals expressing a dominant-negative Rho A for 30 days after resection (*Figure 4F*), injury sites remained deficient in cardiac myofibers, whereas in control hearts had reestablished a contiguous wall of muscle (*Figure 4F*). Proteomic analyses of our whole-cell BioID2 data set also implicated the small GTPases Rac1 and Cdc42 in heart regeneration; although protein levels remained unchanged, pathway analyses showed enrichment for signaling by Rho GTPases and its effectors (*Figure 4D, E*). Similar to Rho A, the interactome maps of Rac1b and Cdc42 revealed many known interactors increased in the whole-cell data set after injury (*Figure 4—figure supplement 1A, B*). Using inducible dominant-negative gene cassettes in transgenic animals, we found that Rac1 and Cdc42 are similarly required for normal heart regeneration (*Figure 4F*, middle and bottom

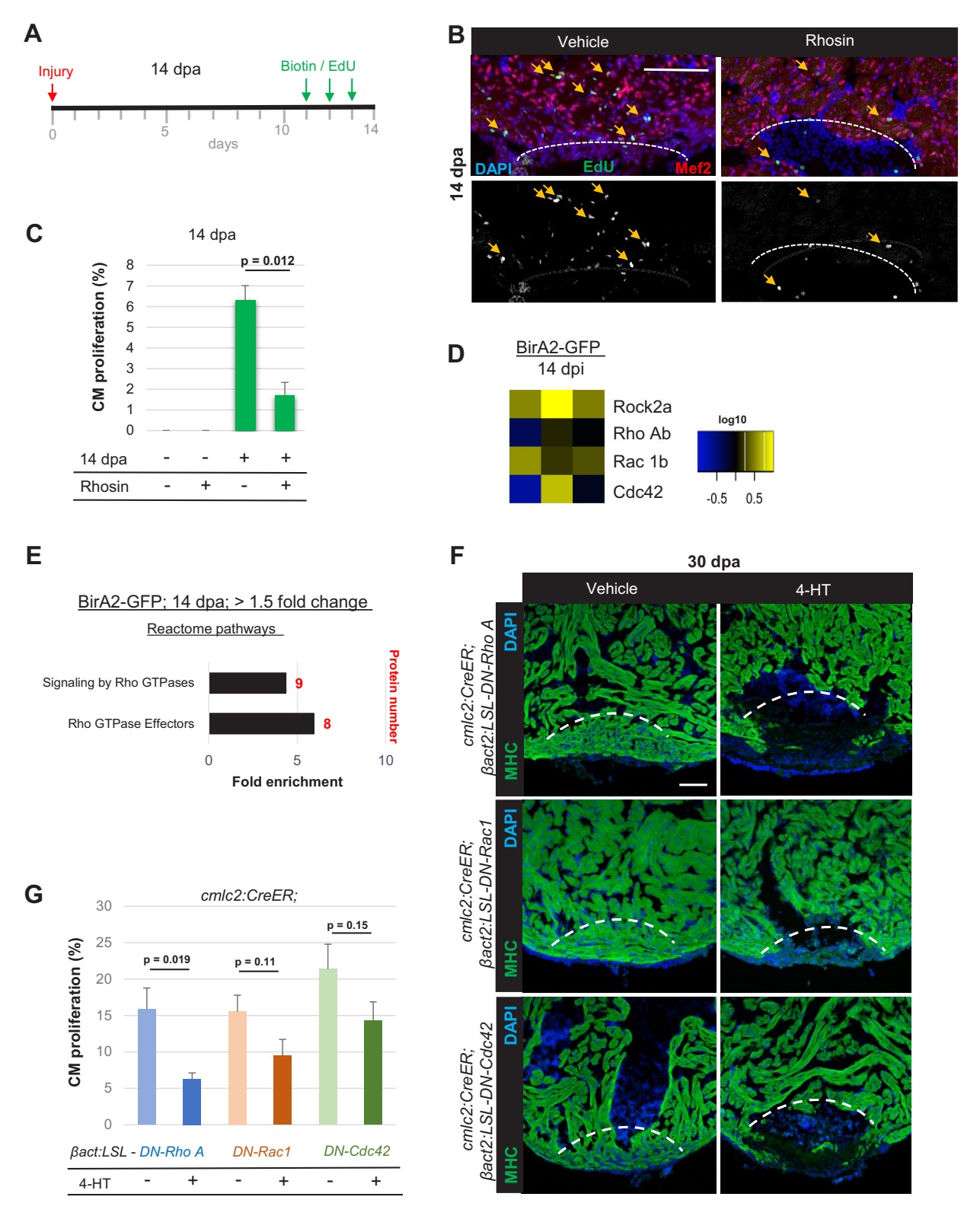

**Figure 4.** Small GTPases are required for cardiac regeneration. (**A**) Timeline of experiment for inhibiting Rho A function in 14 dpa hearts. (**B**) Immunofluorescence images of sections of 14 dpa ventricles vehicle or Rhosin treated and stained for EdU incorporation, an indicator of cardiomyocyte proliferation. Mef2 staining marks cardiomyocyte nuclei. Dashed line, approximate resection plane. Arrowheads, Mef2+/EdU+ cardiomyocytes. (**C**) Quantification of Mef2/EdU assays. Inhibition of Rho A by Rhosin reduces cardiomyocyte (CM) proliferation. n = 5 animals for each condition, two

*Figure 4 continued on next page*

*Figure 4 continued*

independent experiments. Data show mean ± SEM. (Mann–Whitney U test). (**D**) Heat map of indicated proteins from the whole-cell BioID2 data set. Of these, only levels of ROCK2a changed consistently (5.1-fold; p<0.05) during regeneration. (**E**) Over-representation test of reactome pathways. Signaling and effectors of Rho GTPases were found to be over-represented. Fold enrichment as indicated, and protein number in red. p<0.001, FDR < 0.02%. (**F**) Myosin heavy chain (MHC) (green) staining ventricles from animals with induced myocardial-specific dominant-negative Rac, Rho, or Cdc42, along with vehicle-treated ventricles, at 30 dpa. Five to eight animals were assessed in each group treated with vehicle, with none of these animals displaying regeneration defects. Four of five ventricles with induced dominant-negative Rac1, and all ventricles with induced dominant-negative Rho A (n = 7/7), or Cdc42 (n = 9/9), showed obvious areas of missing myocardium. Fisher–Irwin exact test, p<0.05. Dashed line, approximate resection plane. Scale bars, 50 µm. (**G**) Quantification of cardiomyocyte proliferation by Mef2/Proliferating cell nuclear antigen (PCNA) staining of n = 4 animals for each condition. Data show mean ± SEM. (Mann–Whitney U test).

The online version of this article includes the following figure supplement(s) for figure 4:

**Figure supplement 1.** Interactome map of Rac1 and Cdc42.

panels). We also observed tendencies for lowered cardiomyocyte proliferation in each transgenic background at 7 days after resection of the ventricular apex (*Figure 4G*). Thus, our data implicate each member of the small GTPase trio – Rho A, Rac1, and Cdc42 – in zebrafish heart regeneration.

To further test whether Rho A function is required for the effects of ErbB2 signaling, we used transgenic fish enabling tamoxifen-inducible overexpression of Nrg1 ligand in cardiomyocytes (*Figure 5A*; *Gemberling et al., 2015*). Depending on the duration of Nrg1 overexpression, these animals display overt cardiomyocyte hyperplasia and thickening of the ventricular wall (*Figure 5C*; *Gemberling et al., 2015*). We examined Rho A levels from hearts that ectopically expressed Nrg1 for 14 days, and unlike during ablation-induced heart regeneration where Rho A levels are unchanged, we found an ~85% increase in total Rho A protein (*Figure 5—figure supplement 1A,mB*). Consistent with a mitogenic role, three daily injections of zebrafish with Rhosin reduced the cardiomyocyte proliferation index by ~71% in the presence of ectopic Nrg1 expression (*Figure 5B–D*). To examine whether Rho A is key for the activity of other cardiomyocyte mitogens, we tested Rho A levels and the effects of its inhibition during tamoxifen-induced myocardial *vegfaa* overexpression or cardiomyocyte proliferation induced by the vitamin D analog alfacalcidol (*Figure 5A, B*). Western blot analysis revealed that *vegfaa*-expressing hearts had elevated cardiac Rho A levels, whereas alfacalcidol-treated hearts displayed no significant change (*Figure 5—figure supplement 1A, B*). Inhibition of Rho A by Rhosin reduced cardiomyocyte proliferation by ~68% and ~65% during *vegfaa* overexpression and alfacalcidol treatment, respectively (*Figure 5E–H*). Thus, our data indicate that Rho A signaling is a major target during heart regeneration, required for normal cardiomyocyte proliferation in response to injury or presence of Nrg1, Vegfaa, or vitamin D.

Assessment of published single-cell RNA-seq data revealed detectable *rhoaa* expression in cardiomyocytes, endothelial cells, fibroblasts, and immune cells, and *rhoab* expression in cardiomyocytes and endothelial cells during zebrafish heart regeneration (*Supplementary file 6*; *Honkoop et al., 2019*). Additionally, published profiles of cardiomyocyte H3.3 occupancy (*Goldman et al., 2017*), an indicator of active gene expression, indicate an approximately twofold increase in H3.3 enrichment in *rhoaa* and *rhoab* gene bodies during heart regeneration (*Supplementary file 6*). Generation of RNA-seq data from ventricles of transgenic animals after 7 days of induced myocardial Nrg1 overexpression (see Materials and methods) revealed that *rhoaa* levels were weakly elevated (1.6-fold), whereas no changes in *rhoab* or *rhoac* were detectable (*Supplementary file 6*). Thus, *rhoaa* and *rhoab* are top candidates for Rho A function in cardiomyocytes, although conditional disruption of the *rhoa* genes in zebrafish is needed to distinguish which member(s) is key.

## Conclusions

Here, we have applied the proximity labeling technique BioID2 to investigate cell-type-specific proteome changes during a key model for innate tissue regeneration, the renewal of cardiomyocytes upon massive injury to the zebrafish heart. We identified changes specific to cardiomyocytes at the levels of whole-cell, membrane compartment, and the molecular interface of a key pro-regenerative factor ErbB2. Our profiling and functional experiments implicate the GTPase Rho A as mediating the effects of Nrg1 and other mitogens in cardiomyocytes during regeneration and provide a resource for which other candidate regenerative effectors can be assessed. We expect that BioID2 and evolved iterations can be effectively applied in myriad ways to monitor proteome changes during

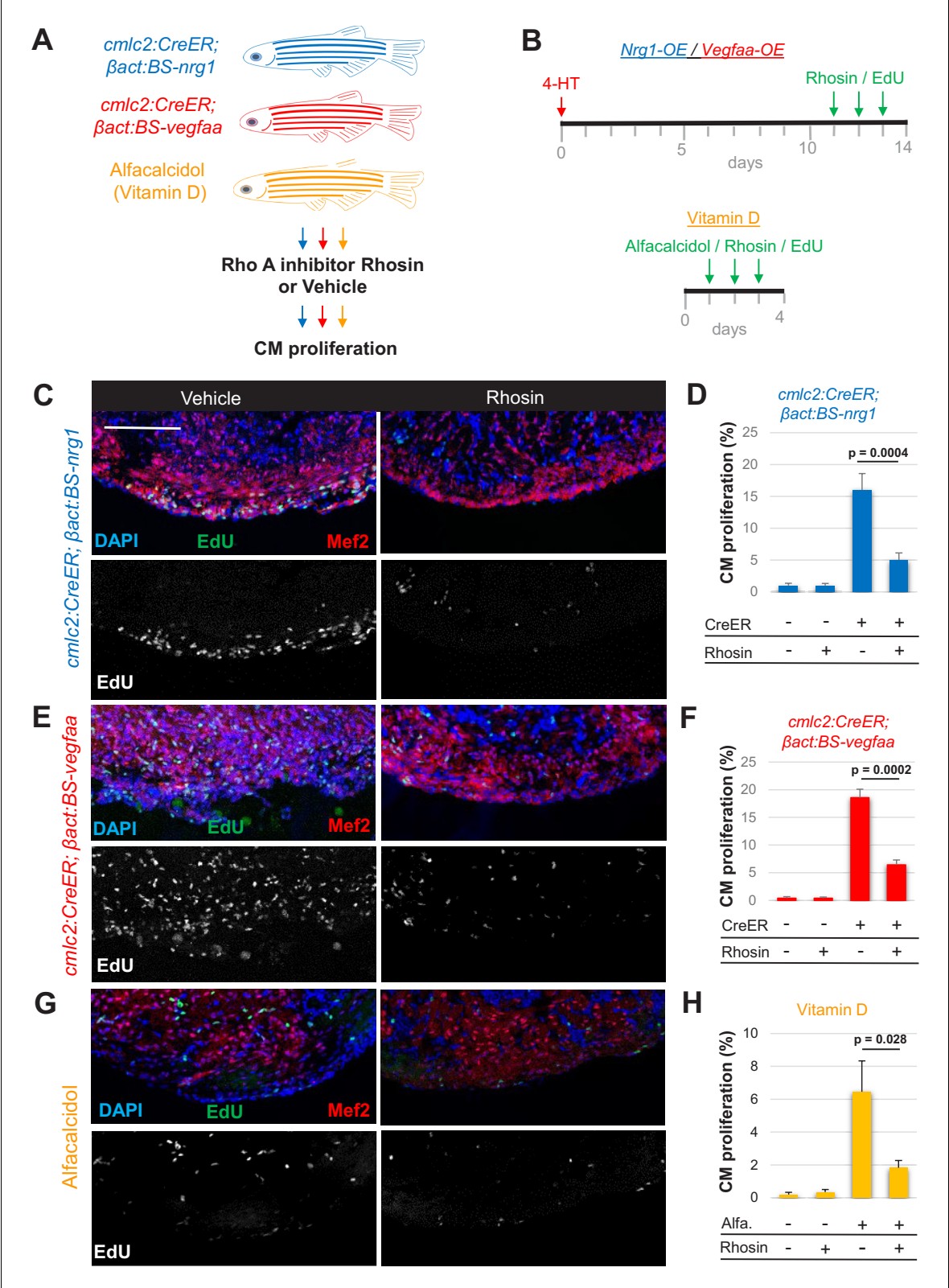

**Figure 5.** Rho A activity is required for cardiogenic responses to mitogens. (**A**) Summary of transgenic animals used in Rho A inhibition experiments. (**B**) Timeline of experiments for Nrg1-overexpression (OE, blue), Vegfaa-OE (red), and alfacalcidol treatment (orange). (**C, E, G**) Immunofluorescence images of ventricles stained for Mef2/EdU from animals overexpressing Nrg1 (**C**) or Vegfaa (**E**) in cardiomyocytes, or injected with alfacalcidol (**G**). Hearts were treated with either vehicle or Rhosin. Scale bar, 50 μm. (**D, F, H**) Inhibition of Rho A by Rhosin reduces cardiomyocyte (CM) proliferation.

*Figure 5 continued on next page*

*Figure 5 continued*

Quantification of Mef2/EdU staining. Five to six animals were assessed for each group in two independent experiments. Data show mean ± SEM (Mann–Whitney U test).

The online version of this article includes the following figure supplement(s) for figure 5:

**Figure supplement 1.** Rho A levels in hearts of cardiac mitogen-treated animals.

dynamic developmental events like regeneration in many species, tissues, and injury contexts to provide high-resolution insights into essential networks of regeneration.

# Materials and methods

**Key resources table**

| Reagent type (species) or resource | Designation | Source or reference | Identifiers | Additional information |
|---|---|---|---|---|
| Genetic reagent (*Danio rerio*) | BirA2-GFP | This paper | *Tg(cmlc2:birA2-GFP)*[pd335] | Transgenic line maintained in K. Poss lab |
| Genetic reagent (*Danio rerio*) | BirA2-GFP-CAAX | This paper | *Tg(cmlc2:birA2-GFP-CAAX)*[pd336] | Transgenic line maintained in K. Poss lab |
| Genetic reagent (*Danio rerio*) | ErbB2-BirA2-HA-P2A-GFP | This paper | *Tg(erbb2-birA2-HA-P2A-GFP)*[pd337] | Transgenic line maintained in K. Poss lab |
| Genetic reagent (*Danio rerio*) | DN-RhoA | This paper | *Tg(βactin2:loxp-BFP-STOP-loxp-DN-Rho)*[pd68] | Transgenic line no longer maintained |
| Genetic reagent (*Danio rerio*) | DN-Rac1 | This paper | *Tg(βactin2:loxp-BFP-STOP-loxp-DN-Rac1)*[pd69] | Transgenic line no longer maintained |
| Genetic reagent (*Danio rerio*) | DN-Cdc42 | This paper | *Tg(βactin2:loxp-BFP-STOP-loxp-DN-Cdc42)*[pd70] | Transgenic line no longer maintained |
| Genetic reagent (*Danio rerio*) | CreER | *Kikuchi et al., 2010* | *Tg(cmlc2:CreER)*[pd10] | Transgenic line, K. Poss lab |
| Genetic reagent (*Danio rerio*) | Nrg1 | *Gemberling et al., 2015* | *Tg(actb2:loxp-TagBFP-STOP-loxp-nrg1)*[pd107] | Transgenic line, K. Poss lab |
| Genetic reagent (*Danio rerio*) | Vegfaa | *Karra et al., 2018* | *Tg(actb2:loxP-TagBFP-loxp-vegfaa)*[pd262] | Transgenic line, K. Poss lab |
| Genetic reagent (*Danio rerio*) | CM ablation | *Wang et al., 2011* | Tg(actb2:loxp-mCherry-loxp-DipTox)[pd36] | Z-CAT cardiac myocyte ablation model, K. Poss lab |
| Antibody | Anti-Mef2 (rabbit polyclonal) | Abcam | Cat#: ab197070 | IF: 1:100 |
| Antibody | Anti-Rho A (rabbit polyclonal) | Santa Cruz | Cat#: 26C4 | IF: 1:1000 |
| Antibody | anti-epb41L5 (rabbit polyclonal) | Proteintech | Cat#: RPL39 | IF: 1:100 |
| Antibody | Anti-Talin 2 (rabbit polyclonal) | Boster Biological Technology | Cat#: PB9961 | IF: 1:100 |
| Antibody | Anti-Seta (rabbit polyclonal) | Antibodies online | Cat#: ABIN2786762 | IF: 1:100 |
| Antibody | Anti-ROCK2 (rabbit polyclonal) | Bioss Antibodies | Cat#: BS-1205R | IF: 1:500 |
| Antibody | Anti-HA (rabbit polyclonal) | Abcam | Cat#: ab9110 | IF: 1:1000 |
| Antibody | Anti-CT3 (mouse monoclonal) | DSHB | Cat#: CT3 | IF: 1:200 |
| Antibody | Anti-PCNA (mouse monoclonal) | Sigma | Cat#: P8825 | IF: 1:200 |

*Continued on next page*

*Continued*

| Reagent type (species) or resource | Designation | Source or reference | Identifiers | Additional information |
|---|---|---|---|---|
| Antibody | Anti-MHC (mouse monoclonal) | DSHB | Cat#: F59-S | IF: 1:200 |
| Chemical compound, drug | Rhosin | Calbiochem | Cat#: 555460 | 250 µM in 10 µl IP injections |
| Chemical compound, drug | Biotin | Sigma-Aldrich | Cat#: B4501 | 0.5 mM in 10 µl IP injections |
| Chemical compound, drug | EDU | Life Technology | Cat#: A10044 | 100 nM in 10 µl IP injections |
| Chemical compound, drug | Alfacalcidol | Selleck Chemicals | Cat#: S1466 | 200 µM in 10 µl IP injections |
| Recombinant DNA reagent | BirA2-GFP | This paper | cmlc2:birA2-GFP | Plasmid used to generate transgenic line |
| Recombinant DNA reagent | BirA2-GFP-CAAX | This paper | cmlc2:birA2-GFP-CAAX | Plasmid used to generate transgenic line |
| Recombinant DNA reagent | ErbB2-BirA2-HA-P2A-GFP | This paper | cmlc2:erbb2-birA2-HA-P2A-GFP | Plasmid used to generate transgenic line |

## Zebrafish

Wild-type or transgenic zebrafish of the hybrid EK/AB strain at 3–8 months of age were used for all experiments. All transgenic strains were analyzed as hemizygotes. Published transgenic strains used in this study were *cmlc2:CreER (Tg(cmlc2:CreER)^pd10^)* (*Kikuchi et al., 2010*), used with *β-act2:BSnrg1* (*Gemberling et al., 2015*), *β-act2:BSvegfaa* (*Karra et al., 2018*), or *β-actin2:loxp-mCherry-STOP-loxp-DTA* (*Wang et al., 2011*). Tamoxifen treatment to deplete cardiomyocytes was performed as described previously (*Wang et al., 2011*) by incubating *cmlc2:CreER; βactin2:loxp-mCherry-STOP-loxp-DTA* (Z-CAT) animals in 0.7 µM tamoxifen (Sigma-Aldrich, St. Louis, MO) for 16 hr. Nrg1 and Vegfaa expression were induced as described by incubating adult *cmlc2:CreER; β-act2:BSNrg1* or *cmlc2:CreER; β-act2:BSVegfaa* animals, respectively, in 5 µM tamoxifen for 24 hr. For expression of DN-Rho A, DN-Rac1 and DN-Cdc42, and Cre-negative animals were bathed in 5 µM 4-HT for 24 hr. For treatment with Rho A inhibitor, 5 µM of a 250 µm Rhosin (*Miranda-Rodríguez et al., 2017*) solution were injected on three consecutive days prior to harvest. Clutch mates were treated with vehicle. EdU injections were performed as described (*Shoffner et al., 2020*). Procedures involving animals were approved by the Institutional Animal Care and Use Committee at Duke University, Protocol number A005-21-01.

Generation of transgenic zebrafish *birA2* (a gift from Scott Soderling) was subcloned into the *cmlc2* vector (*Kikuchi et al., 2010*) into BamHI/NheI sites to generate *cmlc2:birA2*. To generate *cmlc2:birA2-GFP*, an *EGFP* cassette was subcloned downstream of BirA2 using NheI/EcoRI sites. A CAAX-tag was inserted into the EcoRI site to generate *cmlc2:birA2-GFP-CAAX*. *erbb2* cDNA was amplified with the following primers (forward 5′-GCCACCATGGAGGCGGACAGAAGTTT-3′; reverse 5′-TCAGGTGTACTCCTTGTGGCCG-3′) and then subcloned into *cmlc2-birA2* vector using the BsiWI/NruI sites. *HA-P2A-GFP* was then inserted into *cmlc2:erbb2-birA2* by using NheI/XhoI sites to generate *cmlc2:erbb2-birA2-HA-P2A-GFP*. Mouse *GFP-DN-Rho A* (forward 5′-ACCGCCATGGTGAG-CAAGGGCGAAGAG-3′, reverse 5′- TCTGGTTGCCTTGTCTTGTGA-3′), *GFP-DN-Rac1* (forward 5′-ACCGCCATGGTGAGCAAGGGCGAAGAG-3′; reverse 5′- GAAGAGAAAATGCCTGCTGTTGTAA-3′), and *GFP-DN-Cdc42* (forward 5′-ACCGCCATGGTGAGCAAGGGCGAAGAG-3′; reverse 5′-CAACCCAAAAGGAAGTGCTGTATATTCTAA-3′) were a gift from Scott Soderling and were each cloned into *βactin2:loxp-dsRed-STOP-loxp* vector (*Yanazume et al., 2002*) by using the AgeI/NotI sites. Each of these constructs was co-injected with I-Sce-I enzyme mix into one-cell-stage embryos. One founder of each construct was isolated, although they were not maintained as stable lines. The full names of the transgenic lines are as follows: *Tg(cmlc2:birA2-GFP)^pd335^*, *Tg(cmlc2:birA2-GFP-CAAX)^pd336^*, *Tg(erbb2-birA2-HA-P2A-GFP)^pd337^*, *Tg(βactin2:loxp-BFP-STOP-loxp-DN-Rho)^pd68^*, *Tg(βactin2:loxp-BFP-STOP-loxp-DN-Rac)1^pd69^*, and *Tg(βactin2:loxp-BFP-STOP-loxp-DN-Cdc42)^pd70^*.

## BioID2 assay and quantitative mass spectrometry analysis

Twenty hearts from adult zebrafish 3–8 months old were pooled for each sample, and triplicates were used for each condition. Prior to isolation of hearts, 10 µl of a 500 µM biotin/PBS solution was intraperitoneally (IP) injected into adult zebrafish on three consecutive days. Ventricle and outflow tract but not atrium were extracted, rinsed in cold PBS, and collected in ice-cold RIPA buffer. Using an Eppendorf tube pestle, hearts were lysed by 50 strokes, then flash frozen in EtOH/dry ice, incubated on ice for 15 min, and then again homogenized with 50 strokes of an Eppendorf tube pestle. Lysates were then centrifuged at 21,000 rpm for 20 min at 4°C. The supernatant was transferred to a new tube. Meanwhile, NeutrAvidin beads were pre-washed in keratin-free conditions in a laminar flow hood. 25–50 µl of bead slurry were pipetted into a low-protein-binding tube and spun down for 30 s at 2000 rpm at room temperature (RT). Supernatant was removed with a 30 g needle attached to 1 ml syringe. Beads were washed five times with 500 ml RIPA buffer. Next, washed beads were resuspended in 100 µl RIPA buffer and added to each sample. Each tube was sealed with parafilm and incubated overnight with rotation in 4°C cold room. The next day, samples were centrifuged for 1 min at 3000 × g at 4°C to pellet beads, and the supernatant was removed using a 30 g needle. 500 µl 2% SDS in 50 mM ammonium bicarbonate in MS-grade water was added to the bead pellet, and the mixture was transferred into a low-protein-binding tube. Beads were then washed (*Goldman et al., 2017*) twice with 500 µl 0.5% SDS in 50 mM ammonium bicarbonate (*Hoang et al., 2020*), twice with 1% Triton X-100/1% deoxycholate/25 mM LiCl in 50 mM ammonium bicarbonate (*Johnson et al., 2020*), twice with 0.5 M NaCl in 50 mM ammonium bicarbonate, and two times with 50 mM ammonium bicarbonate solution. For each wash step, 500 µl wash solution was used and samples were rocked for 10 min at RT. Samples were then centrifuged at 2000 × g for 30 s at RT, and the supernatant was removed with a 30 g injection needle. To elute the biotinylated proteome, an elution buffer containing 2% SDS, 10% glycerol, 5% β-mercaptoethanol, and 62.5 mM Tris pH 6.8 (NO bromophenol blue) in MS-grade water was prepared and biotin was added freshly to a final concentration of 2.5 mM. 50 µl elution buffer was added to the washed beads. Samples were boiled for 5 min at 95°C on a heat block. During the elution, samples were slowly vortexed three times. Next, samples were centrifuged for 30 s at 2000 × g at RT, and the supernatant was carefully removed using a new 30 g needle attached to a 1 ml syringe and transferred to a new low-protein-binding tube. Samples were stored at −80°C until delivery to the Duke Proteomics and Metabolomics Shared Resource.

Samples were reduced with 10 mM dithiolthreitol for 30 min at 80°C and alkylated with 20 mM iodoacetamide for 30 min at RT. Next, they were supplemented with 15 µl of 20% SDS in 50 mM TEAB, a final concentration of 1.2% phosphoric acid, and 555 µl of S-Trap (Protifi) binding buffer (90% MeOH/100 mM TEAB). Proteins were trapped on the S-Trap, digested using 20 ng/µl sequencing grade trypsin (Promega) for 1 hr at 47C, and eluted using 50 mM TEAB, followed by 0.2% formic acid (FA), and lastly using 50% ACN/0.2% FA. All samples were then lyophilized and resuspended in 12 µl 1% Trifluoroacetic acid (TFA)/2% acetonitrile containing 12.5 fmol/µl yeast alcohol dehydrogenase (ADH_YEAST). From each sample, 3 µl was removed to create a QC Pool sample, which was run periodically throughout the acquisition period. Quantitative LC/MS/MS was performed on 3 µl of each sample using a nanoAcquity UPLC system (Waters Corp.) coupled to a Thermo Fusion Lumos high-resolution accurate mass tandem mass spectrometer (Thermo) via a nanoelectrospray ionization source. Briefly, the sample was first trapped on a Symmetry C18 20 mm × 180 µm trapping column (5 µl/min at 99.9/0.1 v/v water/acetonitrile), after which the analytical separation was performed using a 1.8 µm Acquity HSS T3 C18 75 µm × 250 mm column (Waters Corp.) with a 90 min linear gradient of 5–30% acetonitrile with 0.1% formic acid at a flow rate of 400 nl/min with a column temperature of 55°C. Data collection on the Fusion Lumos mass spectrometer was performed in a data-dependent acquisition mode of acquisition with an r = 120,000,000 (at m/z 200) full MS scan from m/z 375–1500 and a target Automatic Gain Control (AGC) value of 2e5 ions with a 2 s cycle time. Ion trap MS/MS scans were acquired with a Rapid scan rate, 100 ms max injection time, and a target AGC value of 5e3 ions. A 20 s dynamic exclusion was employed to increase depth of coverage. The total analysis cycle time for each sample injection was approximately 2 hr.

Following UPLC-MS/MS analyses, data were imported into Proteome Discoverer 2.2 (Thermo Scientific Inc), and analyses were aligned based on the accurate mass and retention time of detected ions ('features') using Minora Feature Detector algorithm in Proteome Discoverer. Relative peptide

abundance was calculated based on area under the curve of the selected ion chromatograms of the aligned features across all runs. The MS/MS data was searched against the TrEMBL *D. rerio* database (downloaded in May 2018) with additional proteins, including yeast ADH1, bovine serum albumin (BSA), as well as an equal number of reversed-sequence ('decoys') false discovery rate determination. Mascot Distiller and Mascot Server (v 2.5, Matrix Sciences) were utilized to produce fragment ion spectra and perform the database searches. Database search parameters included fixed modification on Cys (carbamidomethyl) and variable modifications on Meth (oxidation) and Asn and Gln (deamidation). Peptide Validator and Protein FDR Validator nodes in Proteome Discoverer were used to annotate the data at a maximum 1% protein false discovery rate.

Relative peptide abundance of triplicates was averaged for each protein, and proteins were confirmed as positive with p<0.05. As biotinylation of proteins depends on lysine availability at the contact site, proteins with one unique peptide were included in the data if relative abundance fit the outlined criteria above. Injured hearts were normalized to uninjured hearts, and a fold change of 1.5-fold was used as a threshold for protein level change. For the membrane proteome, uninjured BirA2-GFP-CAAX-expressing hearts were normalized to BirA2-GFP hearts, and proteins with a 2.5-fold change were considered to be membrane-specific.

Cytoscape was used to create interactome maps. BioGRID and Scaffold were used to search for interactors of ErbB2, Rho A, Rac1b, and Cdc42. Panther gene list analysis was used for over-representation tests in cellular components, signaling pathways and reactome analysis of the BioID2 data sets.

BioID2 raw MS proteomics data sets have been deposited to MassIVE with the data identifier MSV000087028 and are also summarized in *Supplementary files 7* and *8*.

## RNA-seq

Hearts were removed and placed in Hanks Buffered Saline Solution supplemented with 20 U/ml heparin. Outflow tracts and atria were removed, and ventricles were pulled apart with tweezers to remove blood from the lumen. Washed ventricles were then placed in 1 ml Trizol and processed to recover RNA. Further enrichment for mRNA was performed by polyA isolation, and mRNA was processed into RNA-seq libraries for sequencing on an Illumina Hi-Seq 2000. RNA-seq data was deposited at GEO with the data identifier GSE168371.

## Immunofluorescence and imaging

Primary and secondary antibody staining was performed as described (*Kikuchi et al., 2011*). Mef2/PCNA staining has been previously described (*Kikuchi et al., 2010*). EdU/Mef2 staining of unfixed sections of ventricles was performed as follows: fish were injected intraperitoneally with 10 mM EdU on three consecutive days prior to harvest. For tissue preparation, hearts were removed from animals and placed into ice-cold 30% sucrose/PBS, before mounting. Hearts were flash-frozen in EtOH/dry Ice and stored in −80°C. Cryosections were performed to generate 10-μm-thick sections, and slides were air-dried for 30 min and then stored at −20°C. For EdU/Mef2 staining, slides were incubated on a slide warmer for 10 min and then fixed in 3.7% formaldehyde/PBS at RT for 15 min. Next, slides were washed three times 5 min in PBS. EdU staining was performed by mixing 600 μl 1M Tris pH 8.5, 6 μl 1< $CuSO_4$, 6 μl 10 mM azide, and 600 μl 0.5 M ascorbic acid (made fresh). Slides were incubated for 30 min at RT. Next, washes were performed in 2% Triton-X-100/PBS four times each 5 min at RT, and slides were blocked for 1 hr at RT in 2% Triton-X-100/PBS, 5% goat serum, and 1% BSA. Primary antibody anti-Mef2 (Abcam, ab197070) was incubated 1:100 in 2% Triton-X-100/PBS and 5% goat serum overnight at 4°C. The next day, slides were washed in 2% Triton-X-100/PBS four times each for 5 min at RT and stained with the secondary antibody for 2 hr at RT. Lastly. four 5 min washes in 2% Triton-X-100/PBS were performed, and slides were mounted in Vectashield hard set DAPI reagent.

A Zeiss 700 confocal microscope was used to image slides. For EdU/Mef2 and Mef2/PCNA staining, images were acquired of the three largest sections from each ventricle. Mef2[+] and Mef2[+]/EdU[+] or Mef2[+]/PCNA[+] cells were counted manually using Fij (ImageJ) in a muscle strip 100 μm wide along the ventricular periphery or injury border zone. Values from the sections were averaged to compute a proliferative index for each animal. Four animals were sampled for each condition.

## Biochemical analysis

Co-immunoprecipitations of Erbb2-HA in whole zebrafish hearts were performed as follows. Hearts from 20 *cmlc2:erbb2-birA2-HA-P2A-GFP; cmlc2:CreER; βactin2:loxp-mCherry-STOP-loxp-DTA* animals treated with tamoxifen, or from 20 controls without *cmlc2:CreER*, were extracted 14 days after treatment. Ventricle and outflow tract was isolated, rinsed in cold PBS, and collected in ice-cold RIPA buffer. Using an Eppendorf tube pestle, hearts were lysed by 50 strokes, flash-frozen in EtOH/dry ice, incubated on ice for 15 min, and then again homogenized with 50 stokes of an Eppendorf tube pestle. Lysates were then centrifuged at 21,000 rpm for 20 min at 4℃, and the supernatant was transferred to a new tube. 50 µl were saved as an input control and kept on ice during the entire Co-IP protocol. Heart lysates were first pre-cleared with 20 µl of packed protein A beads (Sigma) per lysate on a rocker or tumbler at 4℃ overnight. The next day, 1 µl of the anti-HA antibody (1 µl IgG rabbit for control) was incubated with 20 µl of protein A beads for 1 hr on a rocker or tumbler at 4℃. Next, antibody-bead mix was added to the pre-cleared heart lysates and incubated for 2 hr at on a rocker or tumbler at 4℃. Samples were three times washed with 500 µl lysis and spun down at 800 rpm for 1 min at 4℃. After the last wash, 50 µl of lysis buffer was added to the samples, and samples were boiled with Laemmli buffer and β-mercaptoethanol at 95℃ for 10 min. Samples were stored at −20℃ before western blot analysis.

For whole heart lysates, the first part of the Co-IP protocol was followed until lysates were centrifuged at 21,000 rpm for 20 min at 4℃. The supernatant was isolated, and a Bradford assay (BioRad) was performed to determine protein concentration. Then, samples were boiled with Laemmli buffer and β-mercaptoethanol at 95℃ for 10 min. 25 µg protein from each whole heart sample was used in western blot analysis.

## Antibodies and reagents

Primary antibodies used in this study: anti-Mef2 (Abcam, ab197070, 1:100), anti-Rho A (Santa Cruz, 26C4,1:1000), anti-epb41L5 (RPL39, Proteintech, 1:100), anti-Talin 2 (Boster Biological Technology, PB9961, 1:100), anti-Seta (Antibodies online, ABIN2786762, 1:100), anti-ROCK2 (Bioss Antibodies, BS-1205R, 1:500), anti-HA (Abcam, ab9110, 1:1000), anti-CT3 (DSHB, 1:200), anti-PCNA (1:200, Sigma, P8825), anti-Mef2 antibody used in PCNA/Mef2 staining (gift from Jingli Cao) and anti-MHC (DSHB, F59-S, 1:200). Secondary antibodies were Alexa Fluor 488 goat anti-rabbit IgG (H + L) (Thermo Fisher Scientific, 1:200), Alexa Fluor 547 goat anti-mouse IgG (H + L) (Thermo Fisher Scientific, 1:200), EdU probes (Thermo Fisher Scientific, E10187), HRP goat anti-mouse IgG (Thermo Fisher Scientific, 31430, 1:50,000), HRP goat anti-rabbit IgG (Thermo Fisher Scientific, 65–6120, 1:50,000), and HRP goat streptavidin IgG (Pierce, 21130, 1:5000). Reagents used in this study were biotin (Sigma-Aldrich, B4501, 0.5 mM), Rhosin (Calbiochem, 555460), alfacalcidol (Selleck Chemicals, S1466, 200 µM), EdU (Life Technology, A10044), protein A sepharose beads (Sigma-Aldrich, P3391), NeutrAvidin Resin (Piere, Thermo Fisher Scientific 29202), and WGA Alexa Fluor 633 conjugate (Invitrogen, W21404, 1:100).

## Acknowledgements

We thank the Duke Zebrafish Shared Resource staff for zebrafish care; Erik Soderblum and the Duke University School of Medicine for the use of the Proteomics and Metabolomics Shared Resource, which provided mass spectrometry service; Scott Soderling for the BirA2 and Rho GTPase constructs; Jianhong Ou for help with comparisons of BioID2 and ChIP-seq data sets; Jingli Cao for reagents; and Frank Conlon, Scott Soderling, Valentina Cigliola, and Ruorong Yan for discussions. MIP was supported by the Life Science Research Foundation (LSRF) – Astellas Pharma Postdoctoral Fellowship. SPS was supported by MISU funding from the FNRS (34772792 – SCHISM). KDP acknowledges research support from NHLBI (R35 HL150713), American Heart Association, and Fondation Leducq.

## Additional information

### Funding

| Funder | Grant reference number | Author |
|---|---|---|
| Life Sciences Research Foundation | | Mira I Pronobis |
| National Heart, Lung, and Blood Institute | R35 HL150713 | Kenneth D Poss |
| American Heart Association | | Kenneth D Poss |
| Fondation Leducq | | Kenneth D Poss |
| Fonds De La Recherche Scientifique - FNRS | 34772792 –SCHISM | Sumeet Pal Singh |

The funders had no role in study design, data collection and interpretation, or the decision to submit the work for publication.

### Author contributions

Mira I Pronobis, Conceptualization, Formal analysis, Funding acquisition, Validation, Investigation, Methodology, Writing - original draft; Susan Zheng, Sumeet Pal Singh, Joseph A Goldman, Investigation; Kenneth D Poss, Conceptualization, Resources, Supervision, Project administration, Writing - review and editing

### Author ORCIDs

Mira I Pronobis (ID) https://orcid.org/0000-0002-9861-6596
Sumeet Pal Singh (ID) http://orcid.org/0000-0002-5154-3318
Joseph A Goldman (ID) https://orcid.org/0000-0002-0800-6189
Kenneth D Poss (ID) https://orcid.org/0000-0002-6743-5709

### Ethics

Animal experimentation: Procedures involving animals were approved by the Institutional Animal Care and Use Committee at Duke University, Protocol number A005-21-01.

### Decision letter and Author response

Decision letter https://doi.org/10.7554/eLife.66079.sa1
Author response https://doi.org/10.7554/eLife.66079.sa2

## Additional files

### Supplementary files

• Supplementary file 1. Normalized levels of BioID2 proteins from uninjured *cmlc2:birA2-GFP-CAAX* hearts. List of proteins from quantitative mass spectrometry analysis of *cmlc2:birA2-GFP-CAAX* ventricles. Uninjured BirA2-GFP-CAAX protein levels were normalized to uninjured BirA2-GFP levels. Data are sorted by fold change. Accession number, gene name, description of gene, total unique peptide count, and fold change with p-values are shown.

• Supplementary file 2. Normalized levels of BioID2 proteins from regenerating *cmlc2:birA2-GFP* hearts. List of proteins from quantitative mass spectrometry analysis of *cmlc2:birA2-GFP* ventricles. Regenerating BirA2-GFP protein levels were normalized to uninjured BirA2-GFP levels. Data are sorted by fold change. Accession number, gene name, description of gene, total unique peptide count, and fold change with p-values are shown.

• Supplementary file 3. Normalized levels of BioID2 proteins from regenerating *cmlc2:birA2-GFP-CAAX* hearts. List of proteins from quantitative mass spectrometry analysis of *cmlc2:birA2-GFP-CAAX* ventricles. Regenerating BirA2-GFP-CAAX protein levels were normalized to uninjured BirA2-

GFP-CAAX levels. Data are sorted by fold change. Accession number, gene name, description of gene, total unique peptide count, and fold change with p-values are shown.

• Supplementary file 4. Normalized levels of BioID2 proteins from regenerating *cmlc2:erbb2-birA2-HA-P2A-GFP* hearts. List of proteins from quantitative mass spectrometry analysis of *cmlc2:erbb2-birA2-HA* ventricles. Regenerating ErbB2-BirA2-HA protein levels were normalized to uninjured ErbB2-BirA2-HA levels. Data are sorted by fold change. Accession number, gene name, description of gene, total unique peptide count, and fold change with p-values are shown.

• Supplementary file 5. ErbB2-associated proteins represented in published transcriptome and epigenome data sets. Association of ErbB2 after injury is color coded as in *Figure 3*, with yellow denoting increased association with ErbB2, green denoting no change in association, and blue denoting decreased association.

• Supplementary file 6. *rhoa* gene expression levels represented in published transcriptome and epigenome data sets after cardiac injury or during Nrg1 overexpression. For Nrg1 experiments, normalized RNA-seq read counts are shown.

• Supplementary file 7. Raw data from BioID2 assays used to generate *Supplementary files 1–3*.

• Supplementary file 8. Raw data from Erbb2-HA-BirA2-P2A-GFP BioID2 assays used to generate *Supplementary file 4*.

• Transparent reporting form

## Data availability

RNA-seq data were deposited at GEO with the data identifier GSE168371. BioID2 raw MS proteomics datasets have been deposited to MassIVE with the data identifier MSV000087028.

The following datasets were generated:

| Author(s) | Year | Dataset title | Dataset URL | Database and Identifier |
|---|---|---|---|---|
| Pronobis MI, Poss KD, Zheng S, Singh SP, Goldman JA | 2021 | RNA-seq from Pronobis et al. | http://www.ncbi.nlm.nih.gov/geo/query/acc.cgi?acc=GSE168371 | NCBI Gene Expression Omnibus, GSE168371 |
| Pronobis MI, Poss KD, Zheng S, Singh SP, Goldman JA | 2021 | MS proteomics from Pronobis et al. | https://massive.ucsd.edu/ProteoSAFe/dataset.jsp?accession=MSV000087028 | MassIVE, MSV000087028 |

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
