## [Decision Letter]

**Acceptance summary:**

The manuscript reports the application of proximity labeling approaches in zebrafish to a specific cell type or its membrane compartment to identify changes in proteome and protein interaction networks during heart regeneration. Using a transgenic approach to express the promiscuous Bir2 biotin ligase in whole cardiomyocytes, their membrane compartment, or fused to the regeneration regulator ErbB2 in zebrafish, the authors define cardiomyocyte-specific proteome changes and protein interactions during heart regeneration. They functionally validate a role of the identified ErbB2-RhoA interaction in regulation of cell proliferation during heart regeneration. The reported methods and data sets will be of interest to those interested in applying proximity labeling techniques to study proteome dynamics and protein interactions in vivo in specific cell types and tissues.

**Decision letter after peer review:**

Thank you for submitting your article "in vivo proximity labeling identifies cardiomyocyte protein networks during zebrafish heart regeneration" for consideration by *eLife*. Your article has been reviewed by 2 peer reviewers, including Lilianna Solnica-Krezel as the Reviewing Editor and Reviewer #1, and the evaluation has been overseen by Didier Stainier as the Senior Editor. The following individual involved in review of your submission has agreed to reveal their identity: Ryan Gray (Reviewer #2).

The manuscript reports the application of proximity labeling approaches in zebrafish to a specific cell type or its membrane compartment to identify changes in proteome and protein interaction networks during heart regeneration. Using a transgenic approach to express the promiscuous Bir2 biotin ligase in whole cardiomyocytes, their membrane compartment, or fused to the regeneration regulator ErbB2 in zebrafish, the authors define cardiomyocyte-specific proteome changes and protein interactions during heart regeneration. They functionally validate a role of the identified ErbB2-RhoA interaction in regulation of cell proliferation during heart regeneration. Overall, the data are properly controlled and analysis of datasets are rigorously analyzed in comparison with previous progress for understanding the transcriptional responses during cardiac regeneration. The reported methods and data sets will be of interest to those interested in applying proximity labeling techniques to study proteome dynamics and protein interactions in vivo in specific cell types and tissues.

Essential Revisions:

1. Page 6 " Quantitative BioID2 analysis captured 1113 unique proteins, of which 343 proteins showed a 2.5-fold enrichment in BirA2-CAAX samples when compared to BirA2-GFP samples ". What do the authors mean as 1113 unique proteins, clearly they overlapped with those detected by BirA2-GFP. Please clarify.

2. Figure 2—figure supplement 1, D panel legend – do different lines show data from three separate experiments?

3. For the proteins that showed increased ErbB2 association during regeneration, it would be important to examine how their level of expression at RNA and protein level (if tools are available) changes during regeneration.

4. The authors conclude that the strongest association was with Rho Ab; can the authors comment on specificity of the antibodies used in IP experiments in Figure 3E? Can they distinguish between different RhoAs in zebrafish?

5. As the authors are using pretty blunt reagents to inhibit RhoAa function (Rhosin and DN-RhoA), which of the genes encoding RhoA is affected is unclear. It will be important to assess whether other rhoa genes (rhoaa, rhoac) are expressed in cardiomyocytes? Is their expression altered during regeneration and in transgenics expressing Nrg1. Given the ease of genome editing, this part of the manuscript would be significantly strengthened by experiments assessing heart regeneration in rhoAa mutant fish, or the authors have to have to include these caveats in their interpretations and conclusions.

6. How was cardiomyocyte proliferation during regeneration affected in the transgenic DN-RhoA fish?

---

## [Author Response]

Essential Revisions:1. Page 6 " Quantitative BioID2 analysis captured 1113 unique proteins, of which 343 proteins showed a 2.5-fold enrichment in BirA2-CAAX samples when compared to BirA2-GFP samples ". What do the authors mean as 1113 unique proteins, clearly they overlapped with those detected by BirA2-GFP. Please clarify.

The 1113 unique proteins refer to individual proteins that have been identified in our BioID2 assays of BirA2-GFP and BirA2-GFP-CAAX samples. We now refer to them as follows on p. 6: “Quantitative BioID2 analysis captured 1113 distinct proteins, of which 343 proteins showed a 2.5-fold enrichment in BirA2-CAAX samples when compared to BirA2-GFP samples (Supplementary File 1)”.

2. Figure 2—figure supplement 1, D panel legend – do different lines show data from three separate experiments?

We added to the figure legend of figure 2—figure supplement 1, and supplement 2 and 3: “Heat map summarizes fold changes measured in 3 separate pooled samples.”

3. For the proteins that showed increased ErbB2 association during regeneration, it would be important to examine how their level of expression at RNA and protein level (if tools are available) changes during regeneration.

To evaluate the protein levels of ErbB2 associated proteins (Figure 3C) we have created a table (Supplementary File 5) listing the protein levels that are found in our whole cell BioID2 samples (BirA2-GFP), and as reported in the proteomic screens of Wang et al. 2013 and Ma et al. 2018. Supplementary File 5 also lists the mRNA levels of ErbB2 associated proteins found in previously published RNA-seq datasets (Kang et al. 2016 and Ben-Yair et al. 2019) and CM-specific ChIP-seq of H3.3 (Goldman et al. 2017). We added the following to the manuscript (p. 10): “Next, we examined the levels of these candidates as reported in published transcriptome and proteome datasets (1, 35, 36, 41, 42), and found that overall expression levels were not predictive of ErbB2 association (Supplementary File 5).”

4. The authors conclude that the strongest association was with Rho Ab; can the authors comment on specificity of the antibodies used in IP experiments in Figure 3E? Can they distinguish between different RhoAs in zebrafish?

Due to high conservation, we do not expect that the antibody we used can distinguish between RhoAa, RhoAb, or RhoAc. We have now included an alignment of the different Rho A proteins and the immunogen used to generate the antibody (Figure 3—figure supplement 2), and we also state in our manuscript on p. 11: “Due to their high conservation, we were unable to identify which of the Rho A proteins, Rho A-a, Rho A-b, or Rho A-c increased its association with ErbB2 (Figure 3—figure supplement 2).”

5. As the authors are using pretty blunt reagents to inhibit RhoAa function (Rhosin and DN-RhoA), which of the genes encoding RhoA is affected is unclear. It will be important to assess whether other rhoa genes (rhoaa, rhoac) are expressed in cardiomyocytes? Is their expression altered during regeneration and in transgenics expressing Nrg1. Given the ease of genome editing, this part of the manuscript would be significantly strengthened by experiments assessing heart regeneration in rhoAa mutant fish, or the authors have to have to include these caveats in their interpretations and conclusions.

We have expanded Supplementary File 6, and it now includes the mRNA levels of *rhoaa*, *rhoab*, and *rhoac* genes reported in single cell RNA-seq data (Honkoop et al., 2019), and new RNA-seq data from collected from zebrafish hearts with induced overexpression of Nrg1. These data indicate that mRNA levels of *rhoaa* increase after Nrg1 overexpression, whereas there is no evidence of changes in *rhoab* and *rhoac*. We summarize these results in our manuscript as follows (p. 13): “Assessment of published single cell RNA-seq data revealed detectable *rhoaa* expression in cardiomyocytes, endothelial cells, fibroblasts, and immune cells, and *rhoab* expression in cardiomyocytes and endothelial cells during zebrafish heart regeneration (33) (Supplementary File 6). Additionally, published profiles of cardiomyocyte H3.3 occupancy (1), an indicator of active gene expression, indicate a ~2-fold increase in H3.3 enrichment in *rhoaa* and *rhoab* gene bodies during heart regeneration (Supplementary File 6). Generation of RNA-seq data from ventricles of transgenic animals after 7 days of induced myocardial Nrg1 overexpression (see Methods) revealed that *rhoaa* levels were weakly elevated (1.6-fold), whereas no changes in *rhoab* or *rhoac* were detectable (Supplementary File 6). Thus, *rhoaa* and *rhoab* are top candidates for Rho A function in cardiomyocytes, although conditional disruption of the *rhoa* genes in zebrafish is needed to distinguish which member(s) is key.”

6. How was cardiomyocyte proliferation during regeneration affected in the transgenic DN-RhoA fish?

We have now performed these analyses and added them to the manuscript. We conducted *Mef2*/PCNA staining on DN-Rho A, DN-Rac1 and DN-Cdc42 transgenic animals 8 days after myocardial induction of each transgene, and 7 days after resection of the ventricular apex. We found trends of reduced cardiomyocyte proliferation in each case and added the following to our manuscript (p. 12): “We also observed tendencies for lowered cardiomyocyte proliferation in each transgenic background at 7 days after resection of the ventricular apex (Figure 4G).